# Impact of Enteric Nervous Cells on Irritable Bowel Syndrome: Potential Treatment Options

**DOI:** 10.3390/microorganisms12102036

**Published:** 2024-10-09

**Authors:** Ploutarchos Pastras, Ioanna Aggeletopoulou, Christos Triantos

**Affiliations:** Division of Gastroenterology, Department of Internal Medicine, University Hospital of Patras, 26504 Patras, Greece; ploutarchosp96@gmail.com (P.P.); chtriantos@upatras.gr (C.T.)

**Keywords:** irritable bowel syndrome, gut microbiota, central nervous system, enteric nervous system, enteric nervous cells, signaling, mechanisms, management

## Abstract

Irritable bowel syndrome (IBS) is a condition that significantly impacts the lifestyle, health, and habits of numerous individuals worldwide. Its diagnosis and classification are based on the Rome criteria, updated periodically to reflect new research findings in this field. IBS can be classified into different types based on symptoms, each with distinct treatment approaches and some differences in their pathophysiology. The exact pathological background of IBS remains unclear, with many aspects still unknown. Recent research developments suggest that disorders in the brain-gut–microbiota axis are key contributors to the symptoms and severity of IBS. The central nervous system (CNS) interacts bidirectionally with intestinal processes within the lumen and the intestinal wall, with the autonomic nervous system, particularly the vagus nerve, playing an important role. However, the enteric nervous system (ENS) is also crucial in the pathophysiological pathway of IBS. The apeline–corticotropin-releasing factor (CRF)–toll-like receptor 4 (TLR4) signaling route via enteric glia and serotonin production in enteroendocrine cells at the enteric barrier are among the most well-understood new findings that affect IBS through the ENS. Additionally, the microbiota regulates neuronal signals, modifying enteric function by altering the number of enteric bacteria and other mechanisms. Given the limited therapeutic options currently available, it is essential to identify new treatment targets, with the brain-gut axis, particularly the enteric nervous system, being a promising focus. This study aims to delineate the molecular mechanisms that induce IBS and to suggest potential targets for future research and treatment of this potentially debilitating disease.

## 1. Introduction

Irritable bowel syndrome (IBS) is a common functional disorder that significantly impacts patients’ health and lifestyle globally [1,2]. IBS pathophysiology involves the gastrointestinal tract, genetic predisposition, psychosocial factors, and the central nervous system (CNS) [3]. The complexity and incomplete understanding of IBS pathophysiology have hindered the development of targeted treatments. Recent strategies have shifted from symptom-based approaches to those targeting pathophysiological mechanisms [4]. One such strategy focuses on the brain-gut axis, which plays a crucial role in the disorder’s pathway [5,6]. In this bidirectional interaction, the gut microbiota is an integral component, contributing mainly to enteric neurogenesis as part of the microbiota–gut–brain axis [6]. Thus, studying the enteric nervous system (ENS), gut microbiota, and their interactions may enhance our therapeutic options for IBS and other functional disorders [3].

This study aims to highlight the pathophysiological mechanisms of IBS related to the ENS and gut microbiota and to suggest potential treatments for future research. Specifically, it will delve into how the ENS and gut microbiota interact and contribute to the onset and progression of IBS symptoms. By examining the roles of these components in detail, the study seeks to uncover new insights into the underlying causes of IBS. Additionally, it will propose novel therapeutic strategies that target these mechanisms, paving the way for more effective and personalized treatments in the future. This comprehensive approach may lead to significant advancements in managing IBS and improving the quality of life for those affected by this chronic condition.

## 2. Epidemiology and Diagnostic Criteria of IBS

IBS is a prevalent disorder of the gut–brain axis, previously classified as a functional gastrointestinal disorder [7]. It is a chronic condition that can significantly impair a patient’s quality of life [8]. Common symptoms include recurrent abdominal pain and changes in defecation patterns, which can vary widely in both quality and quantity [9]. Due to its symptoms overlapping with other conditions and challenges in diagnosis, the Rome criteria were developed by an international panel of experts in brain–gut interactions for use in clinical practice and research. The most recent version, Rome IV, was established in 2016 to define IBS and other gastrointestinal disorders [10].

According to the Rome IV criteria, IBS is diagnosed based on symptoms that have been present for at least 3 months, with onset at least 6 months prior to diagnosis. The criteria specify that patients must experience recurrent abdominal pain on average at least once per week over the past 3 months, which must be associated with two or more of the following: a change in stool frequency, a change in stool form (appearance), or defecation [10]. Although bloating is a common symptom, it is not required for an IBS diagnosis [10].

Patients are classified based on their predominant stool pattern into one of the following categories: IBS with diarrhea (IBS-D), where more than 25% of bowel movements are Bristol stool form types 1 or 2 and less than 25% are types 6 or 7; IBS with constipation (IBS-C), where more than 25% of bowel movements are Bristol stool form types 6 or 7 and less than 25% are types 1 or 2; IBS with mixed bowel habits (IBS-M), where more than 25% of bowel movements are both types 1 or 2 and types 6 or 7; or IBS unclassified (IBS-U), for patients who meet the criteria for IBS but do not fit into the other subgroups based on Bristol stool form types [11].

Different IBS subtypes require various treatments. Despite the updated criteria, there can be overlap and fluctuation between IBS and other disorders, which can complicate its clinical application [12,13,14]. However, the Rome IV criteria help differentiate IBS from other functional bowel disorders by specifying the frequency of abdominal pain, and they define IBS as a disorder of gut–brain interaction [10].

The epidemiology of IBS, which varies based on the Rome criteria used, shows significant changes with the introduction of new criteria [15]. A recent study indicated a prevalence of approximately 4.4% to 4.8% in Canada, the United Kingdom, and the United States, with a higher incidence in women and individuals under fifty years old, making IBS a common reason for referrals to gastroenterologists [16]. However, the symptoms of IBS often overlap with those of other disorders, leading to discrepancies in epidemiological studies.

Prior to the Rome IV criteria, a systematic review published in 2012, which covered eighty-one countries and included 260,960 individuals, estimated a global prevalence of IBS at approximately 11% [17]. It is important to note the wide variation in prevalence between countries using different Rome criteria. For example, Pakistan reported a prevalence of 35% using the Rome II criteria, while Iran reported just 1.1% using the Rome III criteria [17].

This variability can be attributed to heterogeneity between studies, which may be influenced by cultural differences, demographic characteristics, and methodological approaches. To address these issues, the Rome Foundation conducted a global survey involving 73,000 individuals across thirty-three countries, using both the Rome III and IV criteria. The survey found a global IBS prevalence of 4.1% with the Rome IV criteria, compared to 10.1% with the Rome III criteria [2].

Women are slightly more affected than men, and prevalence tends to decrease modestly with age, being higher among adults aged 18–39 years under both Rome III and IV criteria [18]. The reduction in prevalence observed with the Rome IV criteria reflects its more restrictive and specific diagnostic requirements compared to the Rome III criteria [15]. This difference may influence treatment decisions and the interpretation of results, as patient populations identified using different criteria may present with varying symptoms and severity. Therefore, future research will need to account for these differences and may require larger participant samples to ensure accurate and representative findings [19,20].

## 3. Neurobiological Mechanisms and Structural Brain Changes in IBS

Signals between the CNS and the ENS are influenced by emotional factors (e.g., depression, anxiety), cognitive factors (e.g., attention, beliefs, expectations), and motivational factors. These signals, along with the memories associated with them, contribute to the generation of IBS symptoms [21]. The brain communicates with the gut through various neural, immunological, and hormonal routes via the brain–gut axis. Key components of these routes include the hypothalamic–pituitary–adrenal (HPA) axis, the sympathoadrenal axis, the ANS, comprising the sympathetic and parasympathetic nervous systems, and the monoaminergic pathways that affect spinal cord reflexes and the dorsal horn [21,22].

The anterior cingulate cortex (ACC) and the medial prefrontal cortex (PFC) play significant roles in transmitting signals by sending inputs to the hypothalamus and amygdala. Through multiple mechanisms, the transmitted information modifies spinal reflexes [23]. This pathway modulates homeostatic body states, such as visceral pain and general gastrointestinal function [24,25]. Additionally, the modulation of emotions, including anger, fear, and sadness, impacts gut function via the two branches of the ANS [26,27,28], facilitated by the emotional motor system (EMS), a cortico–limbic–pontine network, determined by the aforementioned pathway [24,25].

The sympathetic nervous system decreases gastrointestinal (GI) motility, transmission, and secretion by modifying cholinergic transmission and stimulating sphincteric contractions in smooth muscle [29,30,31]. It also affects the mucosal immune systems and mucosa–microflora interactions in the gut [32,33,34,35]. Conversely, the parasympathetic nervous system, primarily through the vagus nerve, facilitates the cephalic phase of gastric secretion and encourages the release of peptides and 5-hydroxytryptamine (a serotonin precursor), thereby accelerating gastrointestinal motility [35]. Like the sympathetic nervous system, the parasympathetic nervous system modulates immune cells, activating macrophages as part of the cholinergic anti-inflammatory reflex [36].

Recent research has identified structural brain features in patients with IBS. Activation of the ACC, insula, prefrontal cortex, cerebellum, and thalamus in response to rectal stimulation is higher in IBS patients compared to those without IBS [4,37]. A study comparing female IBS patients to healthy women found that IBS patients had increased gray matter volume and cortical thickness in the primary and secondary somatosensory cortices, as well as in some subcortical regions. However, there were reductions in cortical thickness, surface area, and gray matter volume in the posterior insula and superior frontal gyrus [38]. Additionally, a thicker left primary somatosensory cortex was observed in response to abdominal pain caused by rectal dilation [4,38]. These brain regions, which are involved in IBS symptoms, could be targeted for treatment. Regulating the activity of these cortical areas through various therapies has been shown to improve IBS symptoms [39,40,41].

Furthermore, IBS patients commonly exhibit reduced parasympathetic activity and increased sympathetic nervous system activity [42]. This imbalance in the ANS is modulated by the vagus nerve. Stress can reduce vagal tone, affecting gut motility, sensitivity, peripheral inflammation, and gut permeability. Additionally, the vagus nerve may indirectly sense changes in the gut microenvironment and transmit this information to the brain [43].

## 4. Mechanisms of Enteric Nervous System Function and Serotonin Regulation in IBS

The bidirectional communication of gut signals involves interactions between autonomic reflexes and the ENS [33]. The ENS has been referred to as the “second brain” by researchers such as Furness and Stebbing because it can manage most aspects of digestion and gut defense independently of conscious sensation, allowing the CNS to focus on higher cognitive functions [33,44]. Thus, much of the gastrointestinal tract’s regulatory control occurs locally through the ENS, maintaining gut homeostasis [45,46]. However, gaps remain in our understanding of ENS pathophysiology and function, despite significant research progress [46].

The specific components of the ENS implicated in the pathophysiology of IBS are depicted in Figure 1.

One key communication pathway between the spinal cord and the ENS involves the brain-derived neurotrophic factor-tropomyosin receptor kinase b-protein kinase m zeta (BDNF–TrkB–PKMζ) signaling from the thoracolumbar spinal cord to enteric glial cells in the lamina propria, which affects visceral hypersensitivity [47,48]. An experimental study has shown that enteric inflammation can be reduced when BDNF is administered to glial cells, which helps restore gut function, supports intestinal barrier maintenance, and promotes neuroplasticity [49]. BDNF interacts with TLRs, particularly TLR4, in enteric nervous cells [50,51]. TLR4 is a prominent factor in IBS pathophysiology, with research indicating that CRF–TLR4 proinflammatory cytokine signaling mediates gut barrier disruption and visceral hypersensitivity [52,53]. Nosu et al. observed that apelin activates CRF–TLR4, creating a proinflammatory cytokine signaling cycle in IBS model rats, which is a central pathway in IBS pathology [54]. This interaction with CRF helps explain the link between stress and IBS, as stress increases intestinal permeability [55]. Additionally, mucosal lipopolysaccharides (LPSs) have been found to increase intestinal permeability [56]. The CRF–TLR4 signaling pathway leads to elevated LPS levels and intestinal permeability, contributing to abdominal pain in IBS [57].

Furthermore, communication between the ENS and specialized gastric epithelial cells, such as enteroendocrine cells and enteric glia, plays a crucial role in IBS [58]. These sensory cells detect gut pressure, microbiome composition, gut contents, and hormones released into the lamina propria. These hormones affect nerves and enteric glia through paracrine, afferent, and efferent signaling [59,60,61,62]. Specifically, stimuli from the gastrointestinal tract influence the expression of tryptophan hydroxylase 1 (Tph1), which regulates serotonin (5-HT) production [63,64]. TLR4 and TLR2 receptors, which respond to stimuli from gut microbiota, are involved in serotonin production [65]. This release of serotonin affects various 5-HT receptors on enteric nerves, regulating gut secretion, motility, and defensive responses [66,67].

Different types of IBS show variations in serotonin levels and gastrointestinal motility. IBS-D patients typically have increased serotonin levels, while IBS-C patients have decreased levels. These changes in serotonin metabolism may influence gastrointestinal motility [68,69]. Abnormal colonic transit times are observed in different IBS subtypes: 10%–20% in IBS-C and IBS-M patients, and 25%–45% in IBS-D patients [70,71]. Oro-caecal transit times are similarly affected [72,73]. IBS-C patients exhibit reduced motility, fewer high-amplitude propagating contractions, and delayed transit, whereas IBS-D patients have increased motility, more high-amplitude contractions, and accelerated transit [74,75]. Despite these differences, abnormal colonic transit and evacuation disorders are central to IBS pathophysiology, contributing to symptoms like constipation, bloating, and abdominal pain. Abnormal oro-anal transit time (OATT) correlates with hydrogen and methane concentrations, with more rapid OATT associated with greater abdominal discomfort, rumbling, and nausea [75]. DuPont et al. found delayed gastric emptying in 76% of IBS patients using a wireless pH/pressure recording capsule [76]. Colonic transit time correlates with stool consistency and, to a lesser extent, stool frequency. However, abdominal pain, flatulence, and bloating do not correlate with colonic transit, while abdominal distension is associated with oro-cecal and colonic transit times and inversely with stool consistency [69,70]. Thus, serotonin production affects gastrointestinal motility through its influence on enteric nerves, which may contribute to various IBS symptoms [66]. Additionally, SCFAs, involved in serotonin production, are found at higher levels in IBS patients. PYY, which regulates gut motility and visceral sensitivity by increasing serotonin levels, is localized in endocrine cells [77]. Recent studies also highlight the involvement of mast cells and eosinophils in ENS function and IBS pathology [78,79,80].

## 5. Influence of Gut Microbiota on Enteric Nervous System Function and IBS Pathophysiology

It is well established that the growth and function of the ENS are influenced by the gut microbiota [80,81,82,83]. The relationship between the gut microbiota and the gut-brain axis has been observed in recent pre-clinical studies [51,84,85,86,87]. Muller et al. compared germ-free mice with those raised under specific pathogen-free (SPF) conditions and observed reduced activity in the ileal myenteric plexus neurons of germ-free mice, indicating potential enteric neuronal under-stimulation in the absence of normal gut microbiota [88]. The impact of microbiota on ENS neuroplasticity and neuronal excitability has also been demonstrated in both rats and mice [83,84,85].

In the context of IBS, the gut microbiota acts as a modulator of the microbiota–gut–brain axis, affecting brain behavior, function, and cognition, and influencing symptoms such as abdominal pain [86]. This interaction involves serotonin and TLRs, particularly TLR2 and TLR4 [89,90]. Anitha et al. found that mice lacking TLR4 and those without gut microbes exhibited delayed intestinal transit and destruction of enteric neurons [50]. Metabolomic studies using proton nuclear magnetic resonance spectroscopy and shotgun metagenomic sequencing have linked IBS to alterations in 5-HT [91]. In a study of 142 IBS patients and 120 controls, it was reported that IBS patients had significantly lower gut microbial diversity compared to controls, with decreased microbial diversity associated with IBS [91].

Specifically, increases in Enterobacteria, anaerobes, *Escherichia coli, Ruminococcus gnavus*, and Bacteroides, and decreases in *Lactobacilli*, *Collinsella*, and *Bifidobacteria* were observed in IBS patients compared to controls. In contrast, Bacteroides and *Allisonella* were more prevalent in IBS-M patients [92,93]. In another study comparing fecal samples from IBS-D patients and controls, decreased abundances of *Clostridiaceae, Leuconostocaceae*, *Enterococcaceae*, *Peptostreptococcaceae*, and *Lachnospiraceae* were noted. At the same time, *Sutterellaceae*, *Acidaminococcaceae*, and *Desulfovibrionaceae* were increased [94].

Antibiotic use may also play a role in altering the microbiota and influencing IBS symptoms [95,96]. Additionally, gastrointestinal infections (bacterial, viral, or parasitic) are known to trigger IBS, potentially through changes in the gut microbiome that affect visceral sensation, intestinal permeability, stool consistency, visceral sensitivity, and gastrointestinal motility, key aspects of IBS pathophysiology [97,98,99]. A study using a mouse model demonstrated that bacterial infection increased intestinal permeability, allowing previously tolerated food antigens to trigger a localized immune response in the gastrointestinal tract, leading to histamine release, altered motility, and visceral hypersensitivity [100].

Although the etiology of dysbiosis in IBS is still unclear, recent research has shown that acute gastrointestinal infections are among the strongest risk factors for IBS, leading to post-infection IBS (PI-IBS) [101]. Infections reduce the diversity of the microbiota, altering the balance of gut microorganisms. Parasitic infections lead to the greatest risk of IBS, followed by bacterial infections, and finally viral infections [97].

Dysbiosis may alter the luminal environment by changing the composition of intestinal proteases, bile salts, and bile acids [102]. Proteases act as mediators of visceral hypersensitivity and intestinal barrier dysfunction [102,103]. Additionally, alterations in bile salts can lead to bile acid malabsorption, which may be a cause of diarrhea in PI-IBS [104]. Some of the pathophysiological mechanisms of PI-IBS in humans, such as visceral hypersensitivity and altered colonic motility, have been observed in a rodent model infected with *Trichinella spiralis* [105,106,107].

While a definitive IBS gut microbiome profile has not yet been established, there have been efforts to integrate longitudinal multi-omics analyses of the gut microbiome, host epigenome, and transcriptome in patients with IBS-D and IBS-C [108,109]. These studies aim to identify subtype-specific and symptom-related variations in microbial composition and function. It remains unclear whether these microbial changes are secondary to other factors such as drug use, diet, or altered physiology, including gastrointestinal transit or water content [110].

## 6. Genetic and Epigenetic Factors in IBS

There are indications that genetic background plays a role in the development and progression of IBS, although the evidence is still preliminary. Both genetic and environmental factors contribute to the familial clustering of IBS [110,111]. Various aspects of IBS, including symptom phenotype, barrier function, inflammatory mediators, neurotransmission regulation, ion channels, and bile acid metabolism, have been associated with numerous single nucleotide polymorphisms (SNPs) [110,111]. However, a meta-analysis assessing genes associated with inflammatory mediators found no clear association with most of these genes [112].

Regarding epigenetic changes in IBS, alterations in DNA, such as changes in microRNAs (miRNAs), appear to be correlated with increased visceral sensitivity and permeability [113]. Specifically, the upregulation of TRPV1 is associated with a decrease in miR-199, which contributes to visceral hypersensitivity and heightened visceral pain [114]. Additionally, recent studies have utilized NanoString mRNA measurements to analyze colonic neuroimmune gene expression. The results indicated that the expression of the TRPV1 gene was higher in Gnotobiotic mice from IBS patients and was correlated with both visceral hypersensitivity and anxiety [115].

## 7. Management of IBS and Potential Treatment Options

IBS diagnosis relies on clinical symptoms defined by criteria from various gastroenterology societies, with the Rome IV criteria being the most widely used [10]. Due to the unclear pathophysiology of IBS, treatment options are varied and depend on the type, symptoms, and severity of the condition [3,116]. Initially, patients often opt for lifestyle modifications, dietary changes, and education. If these approaches fail, therapeutic drugs and other non-pharmacological interventions are considered [4]. Recent guidelines emphasize targeting the microbiota–gut–brain axis, especially for moderate to severe IBS cases, due to its significant role in the disease [117,118].

Lifestyle interventions are crucial in IBS management. Stress reduction and dietary modifications play a significant role in symptom management [4]. According to the British Gastroenterological Society guidelines (2021) and the American College of Gastroenterology (ACG) guidelines (2022), dietary counseling should be the first line of treatment. A low FODMAP (fermentable oligosaccharides, disaccharides, monosaccharides, and polyols) diet is highly recommended, supported by recent randomized controlled trials and meta-analyses [56,119,120,121].

Conversely, diets high in fat, alcohol, fizzy drinks, and insoluble fibers should be avoided, as they may exacerbate flatulence and pain [122,123,124]. Further research is needed to explore the relationship between short-chain fatty acids (SCFAs), such as those found in fruits, vegetables, whole grains and legumes, and IBS, given SCFAs’ role in serotonin production [67]. Additionally, the potential link between obesity and IBS warrants investigation due to the role of Peptide YY (PYY) in obesity and serotonin regulation [77].

The Rome IV criteria also recommend brain–gut axis therapies for IBS treatment [125]. These include hypnotherapy, dynamic psychotherapy, and relaxation therapy, which can alleviate abdominal pain and psychological symptoms [126]. Gut-directed hypnotherapy, in particular, can improve gastrointestinal motility and visceral sensitivity, enhancing symptoms and quality of life [127,128]. Hypnotherapy regulates the autonomic nervous system (ANS), especially the vagus nerve, which coordinates gastrointestinal functions [129,130]. In contrast, other stress-modifying therapies, such as selective serotonin reuptake inhibitors (SSRIs), have shown limited effectiveness despite their potential action on the enteric nervous system. [131]. Future research should explore the apelin–corticotropin-releasing factor (CRF)–toll-like receptor 4 (TLR4) signaling pathway, which is directly linked to stress, and the mechanisms of serotonin production in enteroendocrine cells [55,65].

Pharmacological treatments for IBS are tailored to the type and symptoms of the condition [4]. For IBS with diarrhea (IBS-D), rifaximin, eluxadoline, and alosetron are generally recommended despite their potential adverse effects [118]. For IBS with constipation (IBS-C), the ACG guidelines (2022) strongly recommend linaclotide and conditionally recommend tenapanor, plecanatide, tegaserod, and lubiprostone [117]. Antispasmodics and antidepressants are used for pain relief [132,133]. While many drugs have been studied, their effectiveness has often been limited [134,135,136]. The role of probiotics in treatment remains controversial, with strains such as *Lactobacillus rhamnosus*, *Lactobacillus plantarum*, *Saccharomyces boulardii*, and *Bifidobacteria* being considered appropriate [136,137,138,139]. Prebiotics and synbiotics have shown beneficial effects on intestinal microbiota diversity and activity, and fecal microbiota transplantation is emerging as a promising future treatment for IBS [140,141,142].

Potential treatment targets for future research are shown in Table 1.

## 8. Conclusions

IBS significantly impacts the lifestyle and health of millions of people worldwide. The pathophysiology of IBS remains incompletely understood, which means that current treatments focus primarily on alleviating symptoms rather than addressing the underlying causes of the condition. A critical component in the development and progression of IBS is the microbiota–gut–brain axis. This complex system involves communication between the gut microbiota, the ENS, and the CNS.

Enteric nervous cells, particularly enteric glia, play a vital role in this communication network. They interact with the autonomic nervous system and the gut microbiota, influencing gut excitability and visceral hypersensitivity. This interaction can contribute to the manifestation of IBS symptoms, including abdominal pain and altered bowel habits. Given the central role of the microbiota–gut–brain axis in IBS, developing targeted therapies that focus on this communication pathway is crucial.

One promising avenue for targeted treatment involves TLR4, which is implicated in the immune response and microbial sensing within the gut. TLR4 can influence gut inflammation and permeability, contributing to IBS symptoms. Additionally, the production and regulation of serotonin via enteroendocrine cells are integral to gut function and sensitivity. Serotonin plays a key role in modulating gastrointestinal motility and visceral pain, making it a significant target for therapeutic intervention.

Despite these insights, there is still an imperative need for further research to develop and refine treatment options. The main limitation of the current research is that many studies on IBS have been conducted using different classification criteria for defining the disease. This makes understanding the pathophysiological mechanisms more challenging, as some study participants may not actually have IBS, leading to inaccuracies in the conclusions. However, advances in understanding the interactions between gut microbiota, enteric glia, and the autonomic nervous system could lead to novel therapies that offer more effective relief for IBS patients. Research should continue to explore these pathways and mechanisms to improve treatment strategies and patient outcomes.

## Figures and Tables

**Figure 1 microorganisms-12-02036-f001:**
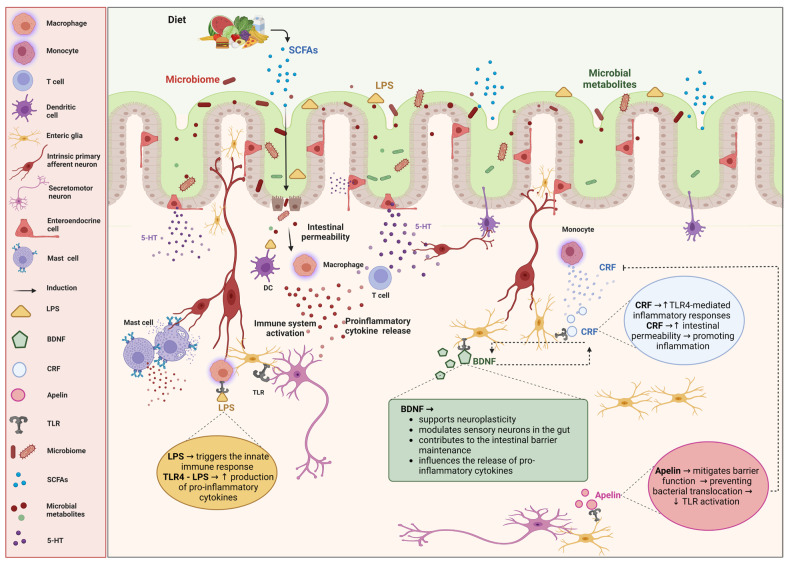
**Impact of enteric nervous cells on irritable bowel syndrome (IBS) and the role of microbiota.** The diagram illustrates the intricate interactions between enteric nervous cells, the gut microbiota, and immune responses, and their impact on irritable bowel syndrome (IBS). The bidirectional relationship between the gut microbiota and the enteric nervous system (ENS) is highlighted, emphasizing how microbial metabolites, immune activation, and neuroimmune interactions contribute to the pathophysiology of IBS. Diet influences the gut microbiome, leading to the production of short-chain fatty acids (SCFAs) and other microbial metabolites, which are crucial for maintaining gut health and modulating the ENS. Alterations in the microbiota can increase intestinal permeability, allowing lipopolysaccharides (LPSs) to cross the epithelial barrier. LPSs activate immune cells such as macrophages, dendritic cells (DCs), T cells, and mast cells, resulting in the release of proinflammatory cytokines. Enteric nervous cells play significant roles in gut physiology and pathology. Serotonin (5-HT), which is released by the enteroendocrine cells, modulates gut motility and sensitivity. Corticotropin-releasing factor (CRF) increases the intestinal permeability and promotes the inflammatory responses through the toll-like receptor 4 (TLR4)-mediated signaling. Brain-derived neurotrophic factor (BDNF) supports neuroplasticity, modulates sensory neurons, and helps maintain the intestinal barrier. Apelin helps mitigate barrier function disruption, preventing bacterial translocation and TLR activation. TLR4 recognizes LPSs, initiating innate immune responses and increasing the production of proinflammatory cytokines. Created with BioRender.com (accessed on 26 August 2024).

**Table 1 microorganisms-12-02036-t001:** Potential treatment targets in the future management of irritable bowel disease.

Apeline–CRF–TLR4 pathway
LPS–TLR4 interaction
TLR4/TLR2 interaction with microbiota
Modulation of pathological microbial diversity in the gut microbiota
Production of 5-HT by enteroendocrine cells
Expression of Tph1 in the gut
BDNF–TrkB–PKMζ signaling
SCFAs
Peptide YY

Abbreviations: CRF, corticotropin-releasing factor; TLR, toll-like receptor; LPS, lipopolysaccharide; 5-HT, serotonin; Tph1, tryptophan hydroxylase; BDNF, brain-derived neurotrophic factor; TrkB, tropomyosin receptor kinase b; PKMζ, protein kinase m zeta; SCFAs, short-chain fatty acids.

## Data Availability

Not applicable.

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
