# Peer review of "Impact of Enteric Nervous Cells on Irritable Bowel Syndrome: Potential Treatment Options"

_microorganisms, 2024, doi:10.3390/microorganisms12102036_

Round 1

Reviewer 1 Report

Comments and Suggestions for Authors

The manuscript titled, “Impact of enteric nervous cells on irritable bowel syndrome: potential treatment options” provided a comprehensive overview of delineating the molecular mechanisms that induce irritable bowel syndrome and providing potential targets for future treatment. Irritable bowel syndrome is a condition that significantly impacts the lifestyle, health, and habits of numerous individuals worldwide. Thus, the biomedical rationale for this review is sound and interest. The major strength of this review is summarizing the current understanding of the pathophysiological mechanisms of irritable bowel syndrome related to the enteric nervous system and gut microbiota, and to suggest potential treatments for future research. This review showed interaction between enteric nervous system and gut microbiota, which contributes to the onset and progression of irritable bowel syndrome symptoms. Overall, this review is straightforward, and the conclusions are supported. However, there are some limitations and concerns, as follows:

Is there any pre-clinical study about irritable bowel syndrome and gut microbiota, or microbiota-gut-brain axis? The authors are suggested to make a table for “potential treatment options”. The limitation of current research should be discussed.

Author Response

Reviewer 1

The manuscript titled, “Impact of enteric nervous cells on irritable bowel syndrome: potential treatment options” provided a comprehensive overview of delineating the molecular mechanisms that induce irritable bowel syndrome and providing potential targets for future treatment. Irritable bowel syndrome is a condition that significantly impacts the lifestyle, health, and habits of numerous individuals worldwide. Thus, the biomedical rationale for this review is sound and interest. The major strength of this review is summarizing the current understanding of the pathophysiological mechanisms of irritable bowel syndrome related to the enteric nervous system and gut microbiota, and to suggest potential treatments for future research. This review showed interaction between enteric nervous system and gut microbiota, which contributes to the onset and progression of irritable bowel syndrome symptoms. Overall, this review is straightforward, and the conclusions are supported. However, there are some limitations and concerns, as follows.

Comment 1: Is there any pre-clinical study about irritable bowel syndrome and gut microbiota, or microbiota-gut-brain axis?

Response to comment 1:  Pre-clinical studies examining the association between the gut microbiota and the microbiota-gut-brain axis are discussed in detail in lines 247-264 and 290-292 of the manuscript. These studies provide insights into the mechanisms underlying the interaction between gut microbial composition and gut-brain signaling in the context of IBS.

Comment 2: The authors are suggested to make a table for “potential treatment options”.

Response to comment 2: A table summarizing the "Potential Treatment Options" has been included in the revised manuscript. 

Comment 3:  The limitation of current research should be discussed.

Response to comment 3: The limitations of current research have been added in lines 396-400 of the manuscript.

Reviewer 2 Report

Comments and Suggestions for Authors

Microorganisms is a specialist journal on microbiota, and the contents of this paper are worthy of publication in the journal. The following revisions are necessary to bring the paper closer to the specialist focus of the journal.

1. The section on Epidemiology and Diagnostic Criteria of IBS should be shortened.

2. The section on Management of IBS should also be shortened and focus on ENS and the gut microbiome.

3. An explanation of the etiology of dysbiosis occurring in IBS is necessary. The relationship between the gut microbiota and the gut-brain axis is also important.

Author Response

Reviewer 2

Microorganisms is a specialist journal on microbiota, and the contents of this paper are worthy of publication in the journal. The following revisions are necessary to bring the paper closer to the specialist focus of the journal.

Comment 1: The section on Epidemiology and Diagnostic Criteria of IBS should be shortened.

Response to comment 1: The section on Epidemiology and Diagnostic Criteria of IBS has been shortened by removing detailed information on diagnostic criteria and epidemiology, as per your suggestion.

Comment 2: The section on Management of IBS should also be shortened and focus on ENS and the gut microbiome.

Response to comment 2: The section on the Management of IBS has been shortened by removing some extraneous information. Treatments related to the ENS have been emphasized, and those associated with the gut microbiota have been highlighted (lines 358-360). Additionally, this section has been repositioned within the manuscript to improve the overall flow, following the incorporation of the reviewer’s comments.

Comment 3: An explanation of the etiology of dysbiosis occurring in IBS is necessary. The relationship between the gut microbiota and the gut-brain axis is also important.

Response to comment 3: The etiology of dysbiosis in IBS is explained in lines 281-292. The relationship between the gut microbiota and the gut-brain axis is highlighted in lines 247-264 and 290-292.